# Global RNA Expression and DNA Methylation Patterns in Primary Anaplastic Thyroid Cancer

**DOI:** 10.3390/cancers12030680

**Published:** 2020-03-13

**Authors:** Naveen Ravi, Minjun Yang, Nektaria Mylona, Johan Wennerberg, Kajsa Paulsson

**Affiliations:** 1Department of Laboratory Medicine, Division of Clinical Genetics, Lund University, SE-221 85 Lund, Sweden; naveen.ravi@med.lu.se (N.R.); minjun.yang@med.lu.se (M.Y.); 2Division of Oncology and Pathology, Clinical Sciences, Lund University and Skåne University Hospital, SE-221 85 Lund, Sweden; nektaria.mylona@skane.se; 3Division of Otorhinolaryngology/Head and Neck Surgery, Clinical Sciences, Lund University and Skåne University Hospital, SE-221 85 Lund, Sweden; johan.wennerberg@med.lu.se

**Keywords:** anaplastic thyroid cancer, DNA methylation, RNA sequencing, formalin-fixed paraffin-embedded tissue

## Abstract

Anaplastic thyroid cancer (ATC) is one of the most malignant tumors, with a median survival of only a few months. The tumorigenic processes of this disease have not yet been completely unraveled. Here, we report an mRNA expression and DNA methylation analysis of fourteen primary ATCs. ATCs clustered separately from normal thyroid tissue in unsupervised analyses, both by RNA expression and by DNA methylation. In expression analysis, enrichment of cell-cycle-related genes as well as downregulation of genes related to thyroid function were seen. Furthermore, ATC displayed a global hypomethylation of the genome but with hypermethylation of CpG islands. Notably, several cancer-related genes displayed a correlation between RNA expression and DNA methylation status, including *MTOR*, *NOTCH1,* and *MAGI1*. Furthermore, *TSHR* and *SLC26A7*, encoding the thyroid-stimulating hormone receptor and an iodine receptor highly expressed in normal thyroid, respectively, displayed low expression as well as aberrant gene body DNA methylation. This study is the largest investigation of global DNA methylation in ATC to date. It shows that aberrant DNA methylation is common in ATC and likely contributes to tumorigenesis in this disease. Future explorations of novel treatments should take this into consideration.

## 1. Introduction

Anaplastic thyroid cancer (ATC), accounting for 1–5% of all thyroid malignancies, is one of the most malignant endocrine tumors, with a patient median survival of less than 1 year [1]. ATC may arise de novo or by dedifferentiation of pre-existing well-differentiated thyroid tumors [2,3]. Deciphering its pathogenesis is likely to lead to the development of more effective therapy. However, the underlying epigenomic and transcriptomic changes that drive ATC are still largely unknown.

Gene expression analyses using microarrays have shown upregulation of genes involved in mitotic cell cycle, epithelial to mesenchymal transition, and TGF-beta signaling in ATC compared with papillary thyroid cancer (PTC) and/or normal thyroid tissue. In contrast, genes related to thyroid hormone synthesis and differentiation have been reported to be downregulated [4,5,6,7,8,9]. However, all expression studies published so far were conducted on small cohorts, including ≤20 cases. Similarly, all DNA methylation studies done so far on ATC have been performed on very small cohorts of samples. Two have included genome-wide methylation analyses using arrays, each including only two or three ATC cases each [10,11]. Putative tumor suppressor genes that have been reported to display promoter hypermethylation in ATC include *PTEN*, *RAP1GAP*, *RASAL1*, *REC8*, *RASSF1*, and *RASSF2* [11,12,13,14,15,16], whereas hypomethylation has been observed in *NOTCH4*, *MAP17*, and *TCL1B*. Notably, *TSHR* and *NKX2-1* (previously *TTF-1*), involved in thyroid functions, have also been reported to be hypermethylated in ATC [17,18]. 

Matched expression and methylation data may unravel genes that are dysregulated by methylation and that could contribute to tumorigenesis. However, no such studies have hitherto been performed on the genomic level in ATC. Here, we applied RNA sequencing (RNA-seq) and genome-wide methylation arrays, including 850,000 CpG sites, on primary ATC tumor samples to identify novel genes that exhibit significant methylation and expression correlation and to further delineate the tumorigenic process of ATC.

## 2. Results

### 2.1. Global Expression Analysis Shows Upregulation of Cell Cycle Genes and Downregulation of Thyroid-Related Genes

Expression levels could be ascertained for 15,043 mRNAs in eleven samples (Appendix A). Unsupervised principal component analysis (PCA) and hierarchical clustering showed clear separate clusters for the ATCs and the normal samples (Figure 1a, Appendix A). In the supervised analysis, 2616 genes were upregulated and 1692 genes were downregulated in ATC compared with normal thyroid tissue. 

To identify up- or downregulated pathways in ATC, enrichment analysis was performed. Pathways that were upregulated in ATC compared with normal thyroid tissue included cell-cycle-related processes, cytokine interactions, extracellular matrix, and G-protein-associated signaling, whereas downregulated pathways were mainly related to translation, transcription, metabolic processes, and mitochondria (Appendix A). Notably, “thyroid hormone generation” and “thyroid hormone metabolic process” were also downregulated. 

To investigate the expression of genes involved in thyroid differentiation, we focused on the genes used to determine the thyroid differentiation score in PTC according to TCGA [19]. Relatively low expression levels of these genes were seen in the majority of cases compared with normal thyroid tissue (Figure 1b), suggesting dedifferentiation, in line with the result from the GSEA. The exceptions were cases 6 and 12, which showed similar expression levels of these genes as in the normal thyroid tissue samples. Furthermore, we ascertained the BRAF-RAS score according to TCGA. This showed that ten of eleven cases were BRAF-like, two of which had *BRAF* V600E mutations (#9 and #11) and one of which had an *NRAS* mutation (#1). One case (#12) was NRAS-like despite not having an *NRAS*-mutation (Figure 1c).

### 2.2. Methylation Analysis Shows Global Hypomethylation and Hypermethylation of CpG Islands

Genome-wide methylation was analyzed in ten primary ATC cases and four normal thyroid tissue samples. These clustered separately in unsupervised PCA and hierarchical clustering analyses (Figure 2a and Appendix A). A total of 13,842 CpG probes were differentially methylated between ATC and normal thyroid tissue, with 1993 probes being hypermethylated and 11,849 being hypomethylated (Figure 2b; Appendix A). Annotation of probes according to their positions showed a slightly higher proportion of hypermethylated probes in gene bodies as well as in shores and CpG islands, whereas more hypomethylated probes were seen in intergenic regions and open sea (Figure 2c,d).

### 2.3. Combined Expression and Methylation Analyses Identify Genes Potentially Involved in Tumorigenesis

To identify genes that were dysregulated by aberrant DNA methylation, we analyzed the seven cases where both RNA expression and methylation data were available (Table 1). We first looked at differentially methylated probes in promoter regions that were associated with a corresponding change in gene expression. We found a total of 191 hypomethylated probes associated with increased gene expression, including *MTOR*, *NOTCH1*, and *HIF1A* (Appendix A). DAVID pathway analysis showed that hypomethylated/overexpressed genes were enriched for cell adhesion, kinases, and cell cycle (Appendix A). Among hypermethylated probes in promoter regions, 30 were associated with decreased gene expression (Appendix A), including *MAGI1*. DAVID pathway analysis showed enrichment for genes involved in cell adhesion and transcription (Appendix A).

Hypermethylation in gene bodies has been reported to be associated with increased gene expression and vice versa [21]. We found a total of 32 genes with hypermethylation in the gene body and significantly increased expression, including *MTOR* (Appendix A). DAVID pathway analysis showed enrichment for cadherins (Appendix A). Furthermore, a total of 226 genes displayed hypomethylation in the gene body and significantly decreased expression, including *TSHR* and *SLC26A7* (Appendix A). DAVID pathway analysis did not show enrichment for any cancer-related processes (Appendix A).

Methylation of regulatory regions may also affect gene expression. To address this, we performed analysis with ELMER [22,23]. A total of 9390 hypomethylated CpG sites in enhancer regions were found to be associated with increased expression of a nearby gene (Appendix A), showing DAVID pathway enrichment for cell-cycle-associated pathways (Appendix A). Of the 289 motifs that were found to be enriched for hypomethylation with ELMER, 13 potential top transcriptional regulators were identified, among them TWIST1 (Appendix A). Furthermore, 476 hypermethylated CpG sites in enhancer regions were found to be associated with decreased expression of a nearby gene (Appendix A) but were not enriched for any cancer-related processes using DAVID (Appendix A). A total of 138 motifs were enriched for hypermethylation with ELMER, including 12 potential top transcriptional regulators (Appendix A).

## 3. Discussion

ATC is a rare disease, and this study is the largest on DNA methylation performed so far as well as the first that has correlated data on gene expression and DNA methylation. Although it remains understudied, the overall genomic landscape of ATC is becoming clearer, and this investigation provides additional clues into the tumorigenic processes in this disease. 

ATC clustered separately from normal thyroid tissue both by expression and by methylation (Figure 1 and Figure 2), in line with previous studies [4,7,10,11] and showing that the tumor cell percentage was relatively high in the tumor samples. The expression signature of ATC was dominated by cell-cycle-related processes (Appendix A), reflecting the high proliferative rate of this malignancy. In regards to DNA methylation, the majority of differentially methylated CpGs in ATC were hypomethylated and in intergenic regions (Figure 2), in line with previous studies [10,11,24]. Global hypomethylation is a common phenomenon in cancer and is believed to be associated with genomic instability [25], agreeing well with the genomic complexity generally seen in ATC [20]. Conversely, most hypermethylated CpG sites were in CpG islands, indicating specific effects on gene expression. 

In addition to the high proliferation, dedifferentiation is a hallmark of ATC. We found that genes associated with thyroid function were enriched in the downregulated fraction of genes in ATC compared with normal thyroid tissue (Appendix A). In line with this, relatively low expression was seen for the thyroid-related genes previously used to determine thyroid differentiation status of PTC for the majority of cases (Figure 1b). The low number of cases in this study prevented analysis of whether low expression of thyroid-related genes correlated with survival, but it can be noted that the two cases (#6 and #12) that retained normal expression of these genes were among the patients that survived the longest (11 and 15 months, respectively; Table 1). Furthermore, we found relatively low expression of *TSHR* and *SLC26A7*, encoding the thyroid-stimulating hormone receptor and an iodine receptor highly expressed in normal thyroid, respectively, as well as aberrant DNA methylation in their gene bodies (Appendix A). Both of these genes have been previously reported to display low expression in ATC, and aberrant methylation has previously been reported for *TSHR* [5,18]. However, a link between expression and methylation has not been previously reported in primary ATC. Taken together, our data suggest that the role of aberrant DNA methylation in ATC dedifferentiation and loss of normal thyroid cell function should be further explored. 

The TCGA study of PTC showed that *BRAF* and *RAS* mutations were highly correlated with different expression patterns, denoted by the BRAF-RAS score [19]. In contrast, Landa et al. [4] reported that most ATCs display a BRAF-like gene expression pattern, regardless of mutational status, a finding recently confirmed by Yoo et al. [26]. In line with these studies, we also found that 10/11 ATC cases in our study displayed a BRAF-like expression pattern, including one case with an *NRAS* mutation (Figure 1c). However, one case (#12) was NRAS-like despite not having a mutation in this gene. This case was also one of two which showed a relatively high thyroid differentiation score (Figure 1b), showing that the correlation between the NRAS-BRAF and thyroid differentiation scores seen in the TCGA study was preserved in our dataset. Notably, we have previously reported that case 12 was also an outlier in terms of mutational pattern, with a very high number of somatic mutations, a different mutational signature, possible microsatellite instability, and occurring in a relatively young patient [20]. 

Aberrant DNA methylation may result in both upregulation of oncogenes and the silencing of tumor suppressor genes. Examples of both of these mechanisms were seen in this study. Oncogenes that displayed both aberrant methylation and increased expression included *MTOR* and *NOTCH1* (Appendix A). *MTOR* encodes a protein kinase promoting cell growth and survival via PI3K/AKT/mTOR signaling [27], and our data suggest that this pathway could also be involved in ATC cases lacking activating mutations in these genes. Considering that clinical trials are ongoing for mTOR inhibitors in ATC [28], the role of epigenetic activation of *MTOR* should be further explored in larger ATC cohorts. NOTCH1 signaling is one of the main pathways involved in cell differentiation, and this gene is frequently dysregulated in cancer [29]. Furthermore, *HIF1A* displayed promoter hypomethylation as well as increased expression in ATC compared with normal thyroid tissue. HIF1A is one of the main players in enabling cells to adapt to hypoxic conditions and has previously been reported to be highly expressed at the protein level in ATC [30,31]. Conversely, *MAGI1* displayed promoter methylation as well as decreased expression in ATC compared with normal thyroid tissue (Appendix A). MAGI1 has been reported to be a tumor suppressor gene in colorectal, gastric, and renal cancer, and knockdown of this gene has been shown to result in migration and invasion in vitro [32,33,34]. Analysis of aberrant methylation in enhancers identified TWIST1 as a potentially important transcription factor. In cancer, TWIST1 has been shown to be involved in epithelial–mesenchymal transition and to promote metastasis [35]. Notably, we have previously reported that *TWIST1* is sometimes amplified in primary ATC (#2 in the present study) [20]. Thus, aberrant activation of TWIST1-associated pathways, either by copy number changes or by aberrant methylation, could be a recurrent event in ATC.

Taken together, we report a high degree of aberrant methylation in ATC, suggesting that epigenetic factors contribute to tumorigenesis. The details of this should be further explored in larger patient cohorts.

## 4. Materials and Methods

### 4.1. Patients

The study included a total of 14 primary cases of ATC previously included in Ravi et al. [20] (Table 1); somatic mutations and gene fusions have been previously reported. Cases were selected based on not having obtained chemotherapy or radiotherapy treatment prior to sampling, sample availability, and >30% tumor cells based on the pathologist’s estimate and/or copy number aberrations or mutations detected by whole-exome sequencing (WES). Thirteen of the samples were formalin-fixed, paraffin-embedded (FFPE) tissue, and one was a fine-needle aspirate obtained at ATC diagnosis. We also included four normal thyroid tissue samples in the study (Table 1). The study was approved by the Ethical Review Board of Lund University (No. 2016/51, 1 February 2016).

### 4.2. Expression Analysis

RNA-seq data from Ravi et al. [20], including 11 ATC cases and four normal thyroid samples (Table 1), were reanalyzed to ascertain mRNA expression. One case (#8) was excluded because of poor sequencing data quality. For #4 and #13, no RNA-seq data was available due to no RNA of sufficiently high quality for RNA-seq being available. Briefly, RNA-seq data were processed using the TCGA mRNA-seq pipeline (https://docs.gdc.cancer.gov/Data/Bioinformatics_Pipelines/Expression_mRNA_Pipeline/#mrna-analysis-pipeline). The sequencing reads were aligned to the human GRCh38 genome assembly using STAR [36], and the read counts for each gene were obtained using HTSeq-count [37]. Genes with count-per-million (CPM) values greater than 1 were defined as expressed genes, and only genes expressed in more than 80% of samples in at least one sample group were used for further analyses. Differential expression analysis was performed using DESeq2 [38], and Benjamini–Hochberg-adjusted (BH-adjusted) *p*-values <0.05 were used as the cutoff for identifying differentially expressed genes. Heatmaps were plotted using GENE-E (https://software.broadinstitute.org/GENE-E/) default settings. Functional enrichment analysis was performed by Gene Set Enrichment Analysis (GSEA) preranked algorithm [39]. The list of genes for the BRAF-RAS score and thyroid differentiation plots were obtained from The Cancer Genome Atlas Research Network (TCGA) study on PTC [19]. Of the 71 BRS genes in the TCGA study, 67 genes were expressed in our dataset, and the BRAF-RAS score was calculated using these according to [19].

### 4.3. Methylation Analysis

The purity of extracted DNA from tumor and normal thyroid tissue was quantitated using NanoDrop (Thermo Fisher Scientific, Waltham, MA, USA), and only DNA with A260/280 >1.7 were included. To assess DNA degradation, all samples were run on agarose gel. In total, 10 tumor samples and four normal thyroid tissue samples could be included in the methylation analysis (Table 1). Cases 5, 6, and 10 were excluded because of too poor DNA quality, and #11 was excluded because no DNA was available. Bisulfite conversion of DNA and methylation profiling using Infinium MethylationEPIC Beadchip arrays (Illumina, Eindhoven, Netherlands) were done according to the manufacturer’s instructions at the Human Genotyping Facility of Erasmus MC (Rotterdam, Netherlands). Methylation analysis was performed using the ChAMP package in R(3.5.2) [40]. Briefly, the raw IDAT files obtained from the array were loaded into ChAMP using the minfi method [41]. Single sample dye correction was performed using the Noob method [42]. Probes with cross-hybridizing potential and polymorphic putative sites with minor allele frequency (MAF) of >5% in the European population were excluded [43]. Probes on sex chromosomes were filtered using ChAMP. The filtered data were normalized using BMIQ in the ChAMP package [44]. Methylation data have been deposited in the GEO database (https://www.ncbi.nlm.nih.gov/geo/) under accession number GSE146003.

### 4.4. Correlation between Methylation and Expression

The normalized methylation BMIQ data were binned into three groups according to β-values: low methylation (≤0.3), moderate methylation (>0.3 but <0.7), and high methylation (≥0.7) [45]. Fisher’s two-sided t-test was performed, with CpG probes with Benjamini–Hochberg-adjusted *p*-values <0.05 and median (Δβ ± 0.3) deemed as statistically significant CpG probes. Filtered CpG probes were annotated as in CpG islands, shores, shelves or open sea as well as annotated as in promoter regions, gene bodies or intergenic regions based on the Infinium MethylationEPIC v1.0 B4 manifest file (http://emea.support.illumina.com/array/array_kits/infinium-methylationepic-beadchip-kit/downloads.html?langsel=/se/). Heatmaps were plotted using GENE-E (https://software.broadinstitute.org/GENE-E/) default settings. Custom Perl scripts were used to correlate CpG sites to expressed genes obtained from DESeq2. Pearson correlations between differentially methylated promoter sites against corresponding fold changes of matched genes were plotted in R. For genes with multiple CpG probes mapping to the promoters, median Δβ value was used.

ELMER (version 2.11.0) supervised analysis model was used to investigate the correlation between gene expression levels and DNA methylation status for enhancers and to identify transcription factor networks regulated by epigenetic modifications [22,23]. Gene expression (FPKM value) and methylation data (BMIQ value) derived from the same sample were used as input (tumor *n* = 7, normal thyroid tissue *n* = 4).

## 5. Conclusions

Aberrant DNA methylation is common in ATC and likely contributes to tumorigenesis. This should be considered in future explorations of novel treatments.

## Figures and Tables

**Figure 1 cancers-12-00680-f001:**
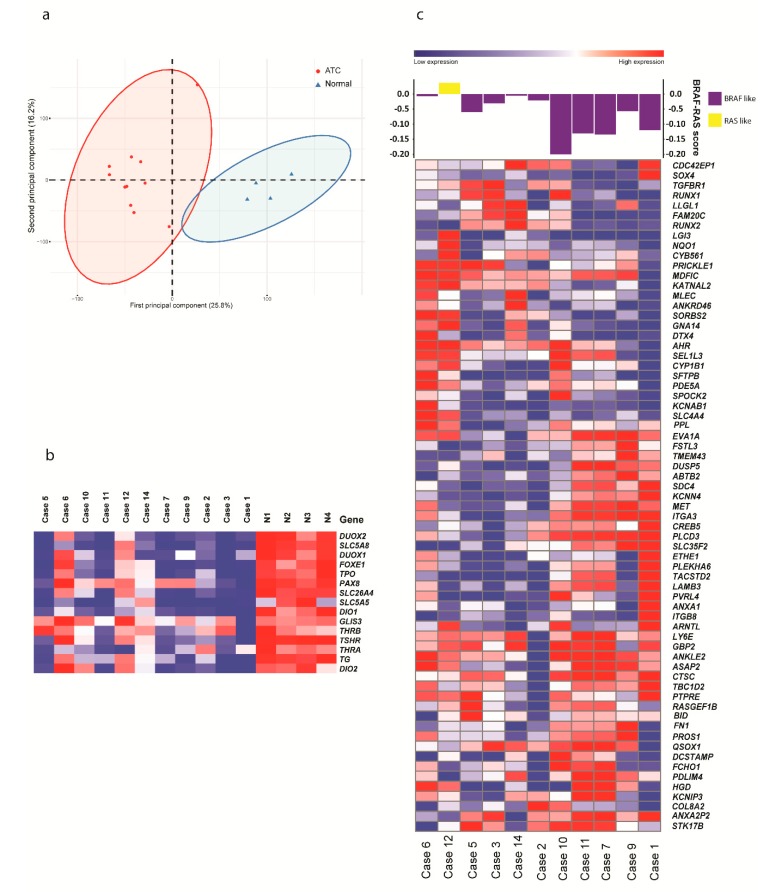
Expression analysis of anaplastic thyroid cancer (ATC). (**a**) Unsupervised clustering by principal component analysis of expression data from 11 ATC cases and tissue from four normal thyroids, showing clear clusters. (**b**) Heatmap displaying relative expression of 15 genes related to thyroid differentiation score in 11 ATC and tissue from four normal thyroids (N1-N4). (**c**) Heatmap displaying expression of 67 genes in BRAF-RAS score signatures in ATC. Based on their expression, cases were classified as BRAF- like (purple) or NRAS-like (yellow) in the top panel.

**Figure 2 cancers-12-00680-f002:**
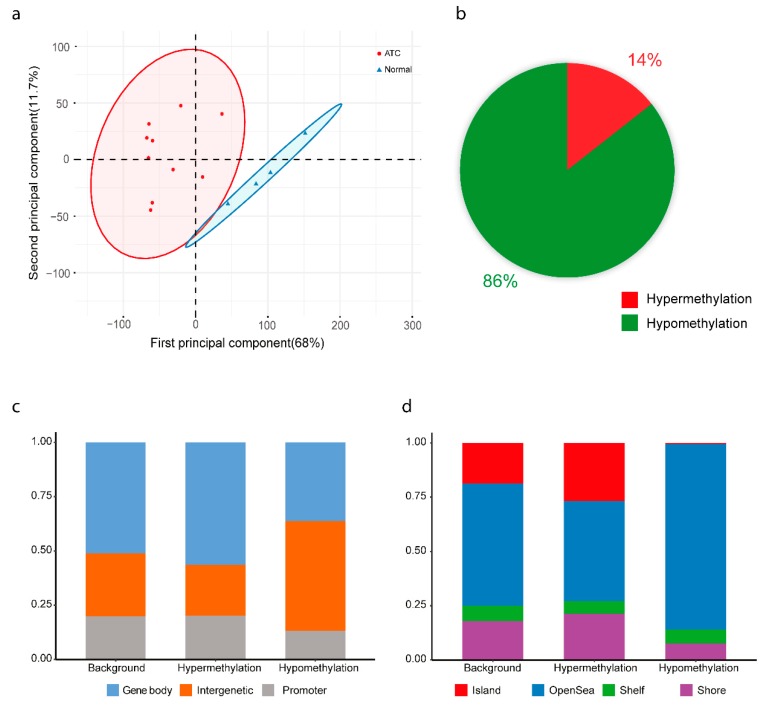
Methylation analysis of anaplastic thyroid cancer (ATC). (**a**) Unsupervised clustering by principal component analysis of methylation data from ten ATC cases and tissue from four normal thyroids. (**b**) Proportions of hypomethylated and hypermethylated differentially methylated probes in ATC. (**c**) Classification of probes based on their location relative to promoter, body and intergenic region based on Illumina EPIC annotation. (**d**) Classification of probes based on location relative to CPG island, shore, shelf, and open sea regions based on Illumina EPIC annotation. Background refers to all the probes in the array.

**Table 1 cancers-12-00680-t001:** Methylation and expression analysis of 14 cases of primary anaplastic thyroid cancer and tissue from 4 normal thyroids (N1-N4).

Case No. *	Gender	Age	Survival (Months)	Expression Analysis	Methylation Analysis
1	F	71	1	Yes	Yes
2	M	70	13	Yes	Yes
3	F	73	8	Yes	Yes
4	M	64	14	No	Yes
5	M	64	4	Yes	No
6	F	72	11	Yes	No
7	F	74	4	Yes	Yes
8	F	84	0	No	Yes
9	F	86	1	Yes	Yes
10	F	70	18	Yes	No
11	M	84	2	Yes	No
12	M	49	15	Yes	Yes
13	M	76	1	No	Yes
14	F	63	7	Yes	Yes
N1	F	62	N/A	Yes	Yes
N2	M	64	N/A	Yes	Yes
N3	F	40	N/A	Yes	Yes
N4	F	56	N/A	Yes	Yes

* Same as Ravi et al. [20]. N/A, not applicable.

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
