# Peer review of "Global RNA Expression and DNA Methylation Patterns in Primary Anaplastic Thyroid Cancer"

_cancers, 2020, doi:10.3390/cancers12030680_

Round 1

Reviewer 1 Report

In this manuscript Ravi and coworkers have analysed mRNA expression and DNA methylation analysis of fourteen primary ATC. Both RNA expression and by DNA methylation of ATC samples clustered separately from normal thyroid tissue.
Cell cycle-related genes resulted enriched whereas downregulation of genes related to thyroid function was observed.
ATC samples displayed a global hypomethylation of the genome and cancer-related genes displayed correlation between RNA expression and DNA methylation status. The B RAF-RAS score showed 10 cases B RAF like status and one N RAS status. In 2 samples the lack of thyroid differentiation genes was not observed.
Similar studies have been previously performed, however in most studies a few ATC samples have been analysed.  The data presented in this study   could be helpful to increase our knowledge about this lesion.
Criticisms
It is not clear how many samples of ATC have been analysed. In the abstract are reported 13 cases , whereas in the main body 14 samples are indicated. In the results section (lanes 6-76) are reported the data obtained in 13 samples. Authors must clarify this point

It is well known that ATC is a rare disease with a very bad prognosis making difficul to obtain samples. The limited number of samples is likely insufficient for statistical studies. Nevertheless authors have not evaluated the prognostic value of their data, at least for the samples showing a different molecular signature ( thyroid differentiation). Authors should give some indications about the prognostic/therapeutic relevance of the different molecular signature observed in the study and discuss about the possible prognostic value of the  their findings.

Author Response

Response to Reviewer 1

We thank the Reviewer for the careful review and constructive criticism.

  • It is not clear how many samples of ATC have been analysed. In the abstract are reported 13 cases , whereas in the main body 14 samples are indicated. In the results section (lanes 6-76) are reported the data obtained in 13 samples. Authors must clarify this point

Response: We apologize for this error. Fourteen samples were included in the study. This has now been corrected throughout the manuscript.

  • It is well known that ATC is a rare disease with a very bad prognosis making difficul to obtain samples. The limited number of samples is likely insufficient for statistical studies. Nevertheless authors have not evaluated the prognostic value of their data, at least for the samples showing a different molecular signature (thyroid differentiation). Authors should give some indications about the prognostic/therapeutic relevance of the different molecular signature observed in the study and discuss about the possible prognostic value of the their findings.

Response: We have now added survival data for all cases to Table 1. However, as the reviewer states, the number of cases are unfortunately too few to make statistical analyses, in particular as the survival times are very short for all patients (range 0-18 months). We have now included a sentence on this in the Discussion (lines 152-156)

Reviewer 2 Report

“Global RNA expression and DNA methylation patterns in primary anaplastic thyroid cancer”

In this manuscript Naveen Ravi and collaborators analyze RNA expression and DNA methylation patterns in a rare type of thyroid tumor. Not only are alterations identified at the transcriptomic and epigenomic level, but they also try to integrate the data obtained from both -omics. The study is interesting and shows potentially valuable data but, in my opinion, several points should be given detailed consideration before publication in Cancers.

Major

1- One of the biggest challenges in the biotechnology and biomedicine fields is the integration of different -omics data. The authors have used a conservative strategy to correlate DNA methylation and expression data, that is, they use only the methylation changes identified in gene promoters, which has traditionally been used to represent the regulatory regions of gene expression. However, the associations between DNA methylation and expression changes are more complex than can be demonstrated in this way. I encourage the authors to use comprehensive analysis tools such as ELMER (Enhancer Linking by Methylation/Expression Relationships), which in addition to integrating methylation and expression data, allows the reconstruction of transcription factor networks, along with the identification of the underlying gene regulatory sequences. What is more, this tool was designed specifically for DNA methylation data obtained with Illumina methylation arrays (450K or 850K). The results of this analysis might improve the quality of the work. Refs. “Yao L, et al. Inferring regulatory element landscapes and transcription factor networks from cancer methylomes. Genome Biol. 2015”; “Silva TC, et al. ELMER v.2: an R/Bioconductor package to reconstruct gene regulatory networks from DNA methylation and transcriptome profiles. Bioinformatics. 2019”.

2- The number of samples used for each analysis should be clearly shown in the text. It is a shame that there are only paired expression and DNA methylation data for 7 samples. To make the results more reliable, DNA methylation and the RNA expression of some candidate CpGs / genes (e.g. top 3/5 candidates) should be validated by bisulfite pyrosequencing and RT-PCR respectively. At the very least, all the samples analyzed in this work should be submitted to these techniques, and if possible, an independent sample cohort.

3- Pag. 4. “Annotation of probes according to their positions showed a higher proportion of hypermethylated probes in promoters and gene bodies as well as in CpG islands, whereas most hypomethylated probes were in intergenic regions (Figure 2C, D)”. This description only indicates the differences in genomic distribution between hyper- and hypomethylated probes, but the enrichment analysis of CpGs should be relativized to the background of the methylation array. The enrichment of the probes in the array is in gene bodies and opensea regions, so the distribution of differentially methylated CpGs could not deviate from what would be expected. In fact, it seems that the distribution of hypermethylated probes does not deviate from what is to be expected in terms of the distribution of all array probes.

Minor

- In abstract and legend of Table 1 it states that 13 tumors are analyzed, although in the table and the methods section (4.1. Patients) there are 14 mentioned. It seems that they are the same 14 patients analyzed at the genomic level by the same authors in Cancer 2019. In addition, 4 cases of normal thyroid are referenced in methods, but no gender or age is described. This information should be added to the text or to Table 1.

- In Figure 1A it seems that only 10 patients are shown, but 11 samples appear in Figures 1B and 1C.

-Supplementary Table 4. It should be indicated in the table legend that the beta values are shown.

-Supplementary Figure 2. The color legend in the figure should indicate “methylation” instead of “expression”. The colors of the Heatmaps of the DNA methylation analyses should be changed to avoid confusion with the Heatmaps of the expression analyses.

Author Response

Response to Reviewer 2

We thank the Reviewer for the careful review and constructive criticism.

  • One of the biggest challenges in the biotechnology and biomedicine fields is the integration of different -omics data. The authors have used a conservative strategy to correlate DNA methylation and expression data, that is, they use only the methylation changes identified in gene promoters, which has traditionally been used to represent the regulatory regions of gene expression. However, the associations between DNA methylation and expression changes are more complex than can be demonstrated in this way. I encourage the authors to use comprehensive analysis tools such as ELMER (Enhancer Linking by Methylation/Expression Relationships), which in addition to integrating methylation and expression data, allows the reconstruction of transcription factor networks, along with the identification of the underlying gene regulatory sequences. What is more, this tool was designed specifically for DNA methylation data obtained with Illumina methylation arrays (450K or 850K). The results of this analysis might improve the quality of the work. Refs. “Yao L, et al. Inferring regulatory element landscapes and transcription factor networks from cancer methylomes. Genome Biol. 2015”; “Silva TC, et al. ELMER v.2: an R/Bioconductor package to reconstruct gene regulatory networks from DNA methylation and transcriptome profiles. Bioinformatics. 2019”.

Response: We thank the reviewer for this very useful suggestion. We have now performed analysis with ELMER and added that to the manuscript, see Results (lines 123-133), Discussion (lines 192-197), Materials and Methods (lines 261-265) and new Supplementary tables 14-19.

  • The number of samples used for each analysis should be clearly shown in the text. It is a shame that there are only paired expression and DNA methylation data for 7 samples. To make the results more reliable, DNA methylation and the RNA expression of some candidate CpGs / genes (e.g. top 3/5 candidates) should be validated by bisulfite pyrosequencing and RT-PCR respectively. At the very least, all the samples analyzed in this work should be submitted to these techniques, and if possible, an independent sample cohort.

Response: We have made sure that the number of samples used for each analysis is included in the Results section. Regarding bisulfite pyrosequencing and RT-PCR, respectfully, we believe that this would be out-of-scope for the current study as this is an initial exploratory analysis. Methylation arrays and RNA-sequencing are robust methods for identifying DNA methylation and mRNA expression, respectively.

  • Annotation of probes according to their positions showed a higher proportion of hypermethylated probes in promoters and gene bodies as well as in CpG islands, whereas most hypomethylated probes were in intergenic regions (Figure 2C, D)”. This description only indicates the differences in genomic distribution between hyper- and hypomethylated probes, but the enrichment analysis of CpGs should be relativized to the background of the methylation array. The enrichment of the probes in the array is in gene bodies and opensea regions, so the distribution of differentially methylated CpGs could not deviate from what would be expected. In fact, it seems that the distribution of hypermethylated probes does not deviate from what is to be expected in terms of the distribution of all array probes.

Response: We thank the reviewer for pointing out this error. We have now redone the analysis including the background level of the probes, see amended Figure 2, Results (lines 94-96) and Discussion (lines 147).

  • In abstract and legend of Table 1 it states that 13 tumors are analyzed, although in the table and the methods section (4.1. Patients) there are 14 mentioned. It seems that they are the same 14 patients analyzed at the genomic level by the same authors in Cancer In addition, 4 cases of normal thyroid are referenced in methods, but no gender or age is described. This information should be added to the text or to Table 1.

Response: We apologize for the confusion regarding the number of cases in the study. Fourteen cases were included; these are the same as in Ravi et al., Cancers 2019. Gender and age for the patients the normal thyroid tissue samples were obtained from have now been added to Table 1.

  • - In Figure 1A it seems that only 10 patients are shown, but 11 samples appear in Figures 1B and 1C.

Response: There are eleven samples shown in Figure 1A. One is high up to the right and two are close together.

  • -Supplementary Table 4. It should be indicated in the table legend that the beta values are shown.

Response: This has been added to the table legends of ST4 and ST5.

  • -Supplementary Figure 2. The color legend in the figure should indicate “methylation” instead of “expression”. The colors of the Heatmaps of the DNA methylation analyses should be changed to avoid confusion with the Heatmaps of the expression analyses.

Response: We have changed the figure color legend and the colors in the heatmap for the DNA methylation analyses.

Reviewer 3 Report

This is a well written manuscript.

Please check whole manuscript for small grammatical errors (eg line 128 As regards should be in regards to etc).

The supplemental tables/figures would need to be slightly more descriptive so that the reader can more easily follow. 

The study needs to include clinical/pathological data from the ATC cases. An analysis would need to be performed to see if the clinical/pathological outcomes correlate with the different tumor types (BRAF-like, NRAS like). Also see next comment.

I suggest that the authors use their data to try to create a prognostic classifier that would be clinically useful. They can their classifier against clinical/pathological data and outcomes.

Why was there a discordance between the cases that the authors performed a methylation analysis and an expression analysis? Since this is small study the authors should mention why on a case by case basis.

Author Response

Response to Reviewer 3

We thank the Reviewer for the careful review and constructive criticism.

  • Please check whole manuscript for small grammatical errors (eg line 128 As regards should be in regards to etc).

Response: We have carefully gone over the manuscript for any grammatical errors.

  • The supplemental tables/figures would need to be slightly more descriptive so that the reader can more easily follow. 

Response: We have gone over the Supplementary Table and Figure legends to make them more descriptive.

  • The study needs to include clinical/pathological data from the ATC cases. An analysis would need to be performed to see if the clinical/pathological outcomes correlate with the different tumor types (BRAF-like, NRAS like). Also see next comment.

Response: Pathological data for all cases is available in Ravi et al., Cancers 2019;11:402. We have now added survival data to Table 1. However, unfortunately the limited number of cases prevents any correlation analysis. We have added a sentence on this in the Discussion (lines 152-156).

  • I suggest that the authors use their data to try to create a prognostic classifier that would be clinically useful. They can their classifier against clinical/pathological data and outcomes.

Response: We agree that a prognostic classifier for ATC would be useful. Unfortunately, we could not do this because of the generally very poor outcome for these patients and the limited number of cases in the present study. However, we have now added survival data to Table 1 and hope that future studies will be able to incorporate these data to build such a classifier.

  • Why was there a discordance between the cases that the authors performed a methylation analysis and an expression analysis? Since this is small study the authors should mention why on a case by case basis.

Response: Cases were excluded based on lack of RNA/DNA of sufficient quality, lack of DNA altogether, and because of poor sequencing data. This has now been added and specified for each case in the Materials and Methods section (lines 213-215 and 235-236).

Round 2

Reviewer 2 Report

The authors have solved the questions and doubts that I had raised. I think this version is now suitable for publication in Cancers.

Reviewer 3 Report

The authors have adequately addressed my comments.